# Synaptic Strength For Convolutional Neural Network

**Chen Lin**
SenseTime Research
linchen@sensetime.com

**Zhao Zhong** *
NLPR, CASIA
University of Chinese Academy of Sciences
zhao.zhong@nlpr.ia.ac.cn

**Wei Wu**
SenseTime Research
wuwei@sensetime.com

**Junjie Yan**
SenseTime Research
yanjunjie@sensetime.com

## Abstract

Convolutional Neural Networks(CNNs) are both computation and memory inten-
sive which hindered their deployment in mobile devices. Inspired by the relevant
concept in neural science literature, we propose Synaptic Pruning: a data-driven
method to prune connections between input and output feature maps with a newly
proposed class of parameters called Synaptic Strength. Synaptic Strength is de-
signed to capture the importance of a connection based on the amount of informa-
tion it transports. Experiment results show the effectiveness of our approach. On
CIFAR-10, we prune connections for various CNN models with up to $96\%$ , which
results in significant size reduction and computation saving. Further evaluation on
ImageNet demonstrates that synaptic pruning is able to discover efficient models
which is competitive to state-of-the-art compact CNNs such as MobileNet-V2 and
NasNet-Mobile. Our contribution is summarized as following: (1) We introduce
Synaptic Strength, a new class of parameters for CNNs to indicate the importance
of each connections. (2) Our approach can prune various CNNs with high com-
pression without compromising accuracy. (3) Further investigation shows, the
proposed Synaptic Strength is a better indicator for kernel pruning compared with
the previous approach in both empirical result and theoretical analysis.

## 1 Introduction

In recent years, Convolutional Neural Networks(CNNs) gradually become dominant in the computer
vision community. Despite their good performance, CNNs have a huge number of parameters,
resulting in high resource demand for storage and computation. Modern CNNs can reach hundreds
of millions of parameters and billions of operations, which makes it difficult to deploy. To alle-
viate aforementioned problem, various methods have been proposed to increase the efficiency of
CNNs. These include knowledge distillation [12, 28], low-rank decomposition [24, 7], network
quantization/binarization [38, 5, 4, 27] and weight pruning [9]. Recent work shows that there exists
large amount of redundancy in CNNs[9, 13]. Accordingly, we can reduce the model size without
compromising accuracy with some appropriate schemes.

Meanwhile, in neural science literature, tremendous redundancy is believed to exist in human
brain [2]. A process of synapse elimination called synaptic pruning removes unnecessary neuronal
structures occurs from birth to adolescence. The pruning process is believed essential to the flexibility
required for the adaptive capabilities of the developing mind [6]. The key element of the brain's
pruning procedure is the synaptic strength. Synaptic pruning in brain follows a "use it or lose it"

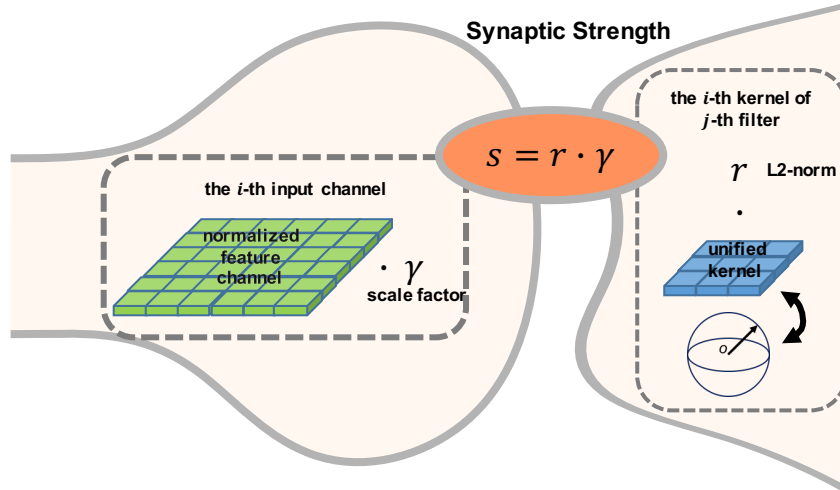

Figure 1: The analogy between neural synapses and CNNs' connections. The normalized and rectified input feature channel(green) is regarded as the information from axon(left). The unified kernel(blue) is regarded as the processing operator on dendrite(right). **Synaptic Strength** is defined as the combined scaling factor which makes it a good indicator for connection importance

principle, which is achieved by increasing synaptic strength if the synapse is used, and the opposite if not [31]. Inspired by this mechanism, we proposed a new class of parameters in convolution layer called **Synaptic Strength** as shown in Figure 1. The main idea of Synaptic Strength is to represent how much information the connection will provide to the final result. This is achieved by imposing normalization on both weight and input data. The analog between biological neural network and CNNs is built by regarding a single channel of the feature map produced by the intermediate convolution layer as a single neuron. Suppose feature channel is produced by a filter contains $C$ kernels and the feature map in the previous layer(also contains $C$ channels). The kernel could be seen as $C$ synapse connected with $C$ previous neurons. Thus the removal of synapse in a biological neural network is analog to disconnect the input channel and its corresponding kernel(s). "Disconnect" can be easily achieved by "zero out" kernel. Thus our synaptic pruning produces kernel-level sparse CNNs.

We evaluate the proposed method on CIFAR-10 and ImageNet. Experiment shows our approach can achieve much higher compression rates while brings less impact on performance compared with existing pruning methods. On CIFAR-10, we can remove up to $96\%$ synaptic connections without decreasing accuracy. On ImageNet, our pruned ResNet-50 achieves competitive or even better efficiency compared with state-of-the-art compact models [29, 39] in term of accuracy and parameters. For compactness, we discuss the practicability of leveraging kernel-level sparsity for acceleration. In particular, we point out that the winograd convolution [18] is ideal to accelerate kernel-level sparse models. We also compare our approach with recent proposed winograd native pruning methods [20, 21] for comparison.

## 2  Related works

**Network Pruning**   The idea of network pruning originated form Yann LeCun and Solla [34]. After deep learning starts to thrive, Han et al. [9] propose to prune the useless weights with small absolute value in model which is trained with L2-regularization and an iterative pruning and finetuning process. Their models achieve high sparse rate at weight-level. They reduce the model size with a large margin while run-time speed up requires specially designed hardware [10]. Recent works [19, 26, 11] focus on the idea of pruning the entire filter. Different filter importance indicator have been proposed. Anwar and Sung [1] first propose to prune parameters at kernel level. They achieve storage saving and acceleration in MRI tasks. Mao et al. [25] apply method proposed in [9] on different granularity to explore the tradeoff between regularity and accuracy. Wen et al. [32] propose to learn a structured sparsity. They use group lasso regularization to prune entire row or column of the weight matrix.

Liu et al. [22] utilize L1 regularization on scale factors in batch-normalization layer to learn the importance indicator for channel pruning. Huang and Wang [15] add sparsity regularization and a modified Accelerated Proximal Gradient on scaling factors in the training stage. They remove unimportant parts of CNN based on scaling factors value. Our synapse pruning is also in an end-to-end manner which is closely related to these ideas. Besides $L_1$ and $L_2$ norm, [23] propose a practical method for $L_0$ norm regularization for CNNs. In order to achieve that, they include a collection of non-negative stochastic gates.

**Neural architecture learning** Optimizing network architecture is explored intensively in the literature. Stanley and Miikkulainen [30] proposed to optimize network typologies and parameter weights at the same time through evolution strategy starting with a minimum neural network. Recently, Zhong et al. [37] and Zoph and Le [39] adopt reinforcement learning for neural architecture search, each of which trains massive amount of different neural networks and treats test accuracy as the reward. Differ with these approaches, our synapse pruning starting with an existing human designed CNN and learning a compact structure through the "use it or lose it" principle. So the proposed method can be regarded as an architecture learning method.

**Irregular connection pattern** Recent proposed efficient CNN architectures aim to reduce both computation and size. These architectures usually conduct special connection patterns between input and out feature channel. Non-dense connection patterns including depth-wise [13, 29], group convolution [33], interleaved group connection[35, 36]. These predefined architecture achieve competitive accuracy with better efficiency. Huang et al. [14] tries to learning the input layers for group convolution in training stage. They impose a constraint on group size. In contrast, synapse pruning has no restriction on the connection pattern at all. We argue that this will give more flexibility to the model.

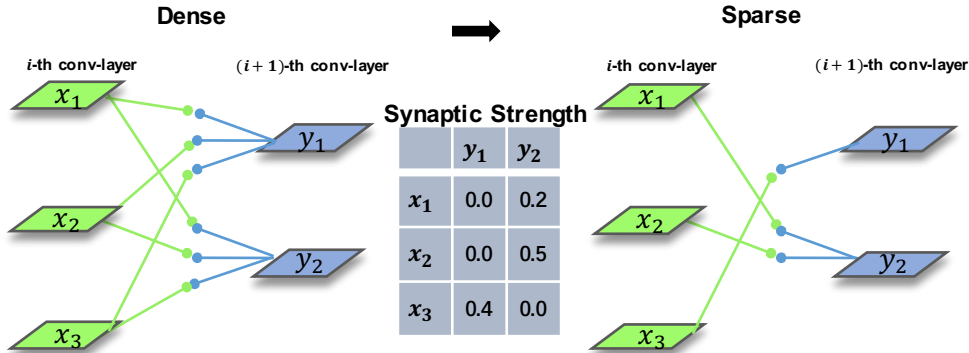

Figure 2: We introduce a new class of parameter called Synaptic Strength associated with connection in convolutional layers (middle). Sparsity regularization is applied on these parameters to automatically identify useless connections. Connections with small Synaptic Strength will be pruned to produce a compact model with kernel-level sparsity.

## 3 Method

### 3.1 Biological Analogy for CNNs

In biology, synaptic strength is a measure of the connectivity between axon and dendrite. Synaptic strength is a changing attribute which represents the activeness of the connection. If a connection is hardly used by the overall processing procedure, synaptic strength is decreased which would finally cause disconnection. In order to adopt the same mechanism for modern CNNs, two question should be answered: (1) How to analog the same mechanism in CNNs. (2) How to define the usefulness of certain connection.

To answer the first question, we provide a demonstration of a convolution layer in Figure 2 (left). This convolution layer takes 3 input feature channel(green). It contains 2 filter, each of which produces a single output feature channel(blue) by adding up the 2D-convolution results produced by each kernel inside the filter and the kernel's corresponding input feature channel. Here, we analogize a single

output feature channel to a biological neuron with 3 dendrites, each of which relates itself with an axon(input feature channel) through 2D-convolution. Kernel for the 2D-convolution decides how the dendrite will process the input. Figure 1 is the close-up version. The "axon"(left) delivers information which is the input feature channel to the receiving "dendrite" on the(right). The "dendrite" perform convolution utilizing the kernel. For the second question, we introduce a new class of parameters to realize the function of synaptic strength.

## 3.2 Define Synaptic Strength for CNNs

Suppose the input feature map contains $C$ channels. The convolution layer has $K$ filters, each of which is consist of $C$ kernels one-to-one assigned to its input feature channel. This convolution operation generates $K$ output feature channels. In general, we have that

$$x_k^{out} = f\left( \sum_{c=1}^{C} x_c^{in} * k_{k,c} + b_k \right) \tag{1}$$

where $f$ represents the activation function and $x_j^{in}$ represents the j-th channel among the input feature map. $k_{k,c}$ represents the $c$-th kernel inside the $k$-th filter. $b_k$ is the bias. Batched Normalization(BN) has been adopted by most modern CNNs as a standard approach for fast convergence and better generalization [16]. We assume the target model perform batch normalization after a convolution layer, before the non-linearity. Particularly, in the training stage, BN layer normalize the activation distribution using mini-batch statistics. Suppose $x_{in}$ and $x_{out}$ is the input and output of a BN layer, $B$ denotes the mini-batch data samples. BN perform normalization by:

$$BN(x) = \frac{x_{in} - \mu_B}{\sqrt{\sigma_B^2 + \epsilon}}; x_{out} = \gamma \cdot BN(x) + \beta \tag{2}$$

where $\mu_B$ and $\sigma_B$ are the mean and standard deviation values computed across each elements in $x_{in}$ over B. The normalized activation $N(x)$ is linear transformed by learnable affine transformation parameterized by $\gamma$ and $\beta$(scale and shift). We also assume that the non-linearity $f$ is homogeneous, which indicates that $f(a \cdot x) = a \cdot f(x)$ for any scalar $a$. The definition of Synaptic Strength could be derived with two modification to the original model. First, we discard channel scaling factor $\gamma$ from BN layers. Second, we reparameterize each individual kernel $k$. We decomposed the kernel as $k = r \cdot k'$, where $r = ||k||$ denotes the Frobenius-norm of the kernel and $k' = \frac{k}{||k||}$ is the normalized(unit) kernel. Utilizing Equation 1 and 2, we show that the modified model has the same representation capacity. Without loss of generality, we consider a sub module consist with "BN-$f$-Conv". Due to the normalization in the BN layer, the bias term in convolution layer is redundant, and thus it has been discarded from the model:

$$x_k^{out} = \sum_{c=1}^{C} f\left( \gamma_c \cdot BN(x_c^{in}) + \beta_c \right) * k_{k,c} \tag{3}$$

$$= \sum_{c=1}^{C} \gamma_c f\left( BN(x_c^{in}) + \frac{\beta_c}{\gamma_c} \right) * (r_{k,c} \cdot k'_{k,c}) \tag{4}$$

$$= \sum_{c=1}^{C} \gamma_c \cdot r_{k,c} \cdot f\left( BN(x_c^{in}) + \frac{\beta_c}{\gamma_c} \right) * k'_{k,c} \tag{5}$$

We define Synaptic Strength as the production of BN's scaling factor and the Frobinus-norm of kernel:

$$s_{k,c} = \gamma_c \cdot r_{k,c} \tag{6}$$

Equation 3 to 5 show the function of our modified model remain identical to the original model.

## 3.3 Intuition and Analysis

The importance of the connection should be measured with how much amount of information it provided. However, estimating information entropy for each dendrite is not efficient. Variance which is also an uncertainty measurement could be regarded as an acceptable compromise. Synaptic

Strength is explicitly designed to be a good indicator to the variance of the intermediate feature produced by a single kernel, which is immediately reduced by summation across $C$ input channels. As shown in Equation 3 to 5, the data distribution coming through BN without the scale factor is convoluted by the kernel. At inference time, BN normalized each data examples using a smoothed version of batch mean and variance, which should be close to the overall statistics for data distribution. By integrating the scale factor into Synaptic Strength, we force the variance of the output of the BN layer stays close to 1. Thus Synaptic Strength controls the variance from data. On the other hand, convolution with kernel certainly affects the variance of the feature. We restrict the kernel to lie on the Unit Sphere. Thus Synaptic Strength parameters are forced to represent the multiplication of data variance and kernel norm which makes it a good indicator of information.

### 3.4 Optimization

In additional to classification loss, we impose sparsity-inducing regularization on the Synaptic Strength $\mathbf{s}$. After training, we prune the synapses whose strength is smaller than threshold $\tau$. Finally, we fine-tune the pruned network. Precisely, the training objective function is given by

$$L = \frac{1}{N} \sum_{i=1}^{N} l(y_i, f(x_i, W)) + \lambda \sum_{s \in \mathcal{S}} g(\mathbf{s}) \qquad (7)$$

where $x_i, y_i$ denote the $i$-th training data and label, $W$ denotes all the trainable weights in the model, $l$ is the classification loss, the second sum-term is the sparse-inducing regularization, and $s$ the Synaptic Strength. $\lambda$ is a scalar factor which controls the scale of the sparsity constraint. We choose $g(s) = |s|$ in our experiments and use sub-gradient descent due to non-smooth point at 0.

## 4 Experiments

**Dataset**  In order to evaluate the effectiveness of synapse pruning, we experiment with CIFAR-10 and ImageNet. CIFAR-10 dataset contains 50,000 train examples and 10,000 test examples. Each example contains a single object draw from 10 classes with resolution $32 \times 32$. In each experiment, we perform standard data augmentation including random flip and random crop. ImageNet dataset is a large-scale image recognition benchmark which contains 1.2 million images for training and 50,000 for validation, each image belongs to one out of the total 1,000 classes. Both top1 and top5 single center crop accuracy is reported.

### 4.1 Training Detail

**Baseline**  We train all models we are going to prune from scratch as the baseline model. All the networks are optimized with Stochastic Gradient Descent(SGD) with momentum 0.9 and weight decay $10^{-4}$. For CIFAR-10 models, we train them with batch size 128 for 240 epochs in total. The initial learning rate is set to 0.1 and divided by 10 at the beginning of 120 and 180 epoch. For ImageNet models, we train them with batch size 256 for 100 epochs in total. The initial learning rate is set to 0.1 and dived ed by 10 at the beginning of 30, 60 and 90 epoch.

**Guidelines for Picking $\lambda$**  For CIFAR-10 models, we pick different sparsity regularization rate $\lambda$ as defined in Section 3.4 for different architectures. We pick $\lambda$ equals to $10^{-4}$ for VGGNet, while $10^{-5}$ and $5 \times 10^{-6}$ for ResNet-18 and DenseNet-40. Other settings remain identical to CIFAR-10 baseline models. For ImageNet models, more flexibility is required for fitting train data. Thus the sparsity constraint rate is set to $10^-6$. The rest of the setting is the same as the baseline routine. Our empirical experience suggests that a higher regularization rate would lead to more pruned kernels with no cost at pruning procedure. But if the regularization is too strong, the accuracy before pruning would be compromised. A basic strategy is to start with a small $\lambda$ and enlarge it until the performance of the model starts to decrease.

**Pruning and finetune**  Since Synaptic strength could represent the information extracted by its owner kernel from the corresponding input channel, we apply a simple pruning strategy which is to remove all the synapse connection under threshold $\mathbf{t}$. The threshold for pruning is decided by the desired sparsity $k\%$. The value of least $k\%$ synaptic strength in the network will become the threshold.

Table 1: Errors and pruning ratio on CIFAR-10

| Model | Error(%) | Kernels | Pruned(%) | Flops | Pruned(%) |
|---|---|---|---|---|---|
| VGG base | 6.66 | 2,224,320 | 0.00 | 398M | 0.00 |
| VGG pruned | **6.23** | 88,972 | 96.00 | 94M | 76.38 |
| ResNet-18 base | 6.45 | 1,392,832 | 0.00 | 555M | 0.00 |
| ResNet-18 pruned A | **5.06** | 113,436 | 90.00 | 137M | 75.32 |
| ResNet-18 pruned B | 5.80 | 64,477 | 95.00 | 88M | 84.14 |
| DenseNet-40 base | 5.24 | 226,728 | 0.00 | 282M | 0.00 |
| DenseNet-40 pruned | 5.58 | 45,345 | 80.00 | 71M | 74.82 |

Empirically, we prune the network using different $k\%$ and choose the best model considering its size and accuracy. We create a mask which indicates remain kernels for simplicity. For deployment, Block Compressed Row Storage could be applied to save memory space. Finetuning should be applied if accuracy drop is observed after pruning, which takes 50 epochs and 20 epochs for CIFAR-10 and ImageNet models respectively.

## 4.2   Results on CIFAR

**Compared to baseline**   The motivation of our synapse pruning is to minimize the computation and storage cost for CNNs. Each of our pruned models could achieve up to $95\%$ sparsity taking 2D-kernel as a unit, while still maintaining similar accuracy compared with baselines. The parameter saving in convolution layers and FLOPs reductions is shown. For VGGNet and ResNet-18 we pruned $96\%$ and $90\%$ of the total synapse with a slight increase in test accuracy. We attribute this to the regularization effect provided by L1 loss. Even for DenseNet-40, a relatively compact model which conduct feature reuse between layers intensively, synapse pruning could still remove a majority of synapses (about $80\%$) with $0.34\%$ loss in accuracy. For computation reduction, synapse pruning is able to reduce up to $80\%$ of FLOPs. Notably, the proportion of FLOPs reduction is less than kernel reduction due to the different kernel and input size. It is possible to adjust $\lambda$ based on the amount of calculation per layer to alleviate the problem. Table 1 shows the resulted models after pruning compared to baseline on CIFAR-10. Figure 3(Left) visualize the amount of connection we pruning relative to the baseline model.

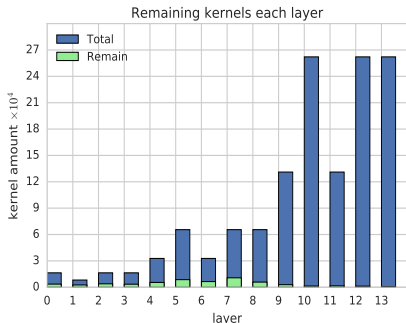

| Model | Error(%) | Paras | Flops |
|---|---|---|---|
| VGG [22] | **6.20** | 2.30M | 195M |
| VGG ours | 6.23 | **0.80M** | **94M** |
| ResNet-56 [19] | 6.94 | 0.73M | 90M |
| ResNet-101 [19] | 6.45 | 1.68M | 213M |
| ResNet-164 [22] | 5.27 | 1.21M | 124M |
| ResNet-18 ours | **5.06** | 1.01M | 137M |
| ResNet-18 ours | 5.56 | **0.49M** | **88M** |
| CondenseNet$^{light}$-94 | **5.00** | 0.33M | 122M |
| DenseNet-40 [22] | 5.19 | 0.66M | 190M |
| DenseNet-40 ours | 5.58 | **0.21M** | **72M** |

Figure 3: (Left) Remaining kernels of ResNet-18 pruned B. (Right) Compare with other compression methods on CIFAR-10

**Compared with other methods**   As shown in Figure 3(Right), compared to state-of-the-art filter-level weight pruning methods, synapse pruning achieves similar accuracy with roughly $1/3$ parameters and at most $1/2$ flops for all three models. Compared to recent proposed compact model CondenseNet, our model could save $30\%$ parameters and $50\%$ computations with an accuracy drop of $0.58\%$. Condensenet [14] also take advantage of irregular connection patterns using a special indexing layer followed by group convolutions, which also introduce extra computation burdens compared with filter-level pruning.

Table 2: Compare with existing pruning based methods for Resnet-50 on ImageNet

| Model | Top1 error(%) | Top5 error(%) | Parameters |
|---|---|---|---|
| ResNet-50 | 24.70 | 7.80 | 25.6M |
| ResNet-50 [15] | $\sim$26.80 | - | $\sim$16.5M |
| ResNet-50 [17] | 31.58 | 11.70 | 8.66M |
| ResNet-50 [25] | - | 7.93 | $\sim$10.22M |
| ResNet-50 ours | **25.32** | **7.20** | **5.9M** |

Table 3: Compare with state-of-the-art compact models on ImageNet

| Model | Top1 error(%) | Top5 error(%) | Parameters |
|---|---|---|---|
| ShuffleNet 2$\times$5.3M | 29.10 | 10.2 | 5.3M |
| CondenseNet | 26.20 | 8.3 | **4.8M** |
| NasNet-A Mobile | 26.00 | 8.4 | 5.3M |
| MobileNet-v2 1.4$\times$ | **25.30** | - | 6.9M |
| ResNet-50 pruned ours | 25.32 | **7.2** | 5.9M |

## 4.3 Results on ImageNet

To further evaluate the performance of synapse pruning on a larger dataset, we perform our method on ImageNet dataset with ResNet-50. Our pruned models removed about $87\%$ connections compared to base models with only $0.6\%$ accuracy drop. As shown in Table 2, we obtain better accuracy with fewer parameters while maintaining the lowest error rate compared to the existing method. Furthermore, we compare the pruned model with state-of-the-art compact models in terms of parameter numbers as shown in Table 3.

## 4.4 Analysis

In this section, we first compare the robustness between the proposed method and SSL[32]. Furthermore, ablation studies by (1)excluding $\gamma$ from Synaptic strength and (2)omitting the kernel reparameterization are performed. Finally, we analysis the effect of hyper-parameter $\lambda$, which is the sparsity regularization rate(see Equation 7). All the experiment is performed with ResNet-18 on CIFAR-10.

**Sensitivity** Wen et al. [32] adopted group-LASSO regularization to push the weight in predefined groups towards zero, which is applicable to kernel pruning. Kernel pruning is based on a global threshold across all layers. Kernels with the mean absolute value less than the threshold are pruned. The motivation of this experiment is to compare the robustness of two approaches at different pruning rate. In order to alleviate the performance gap generated by randomness in training, instead of accuracy, we show the result by plot accuracy drop(caused by pruning) against sparsity. Fine-tuning is performed after each pruning procedure. The results are summarized in Figure 4(a). The sparsity of connections we plotted is $70\%$, $80\%$, $90\%$, $95\%$, $97.5\%$. Our approach starts to outperform SSL from $90\%$ sparsity. From $90\%$ to $97.5\%$, the gap between the two methods become larger. If we constraint the accuracy drop no be less than $1\%$, the proposed method could produce a model with roughly $2\times$ fewer kernels.

**Ablation** There are two modifications to the original model in order to perform synaptic pruning (1) discard the scale factor from previous batch-norm layer $\gamma$ and (2) apply normalization to the kernel and explicitly parameterize kernel's $L2$-norm. We train two variants of synaptic pruning each of which omitted one of the aforementioned modifications to show the necessity. We refer the model trained without discarding the scale factor as "non fix $\gamma$", the other one as "non kernel-norm". The plot sparsity of connections is chosen as $60\%$, $70\%$, $80\%$, $90\%$. From the accuracy-sparsity curve showed in Figure 4(b), the full version of the proposed method gets the highest accuracy when pruning rate greater than $80\%$. Omitting either one of these modification degrades the performance. In order to highlight the disparity, finetuning is not performed.

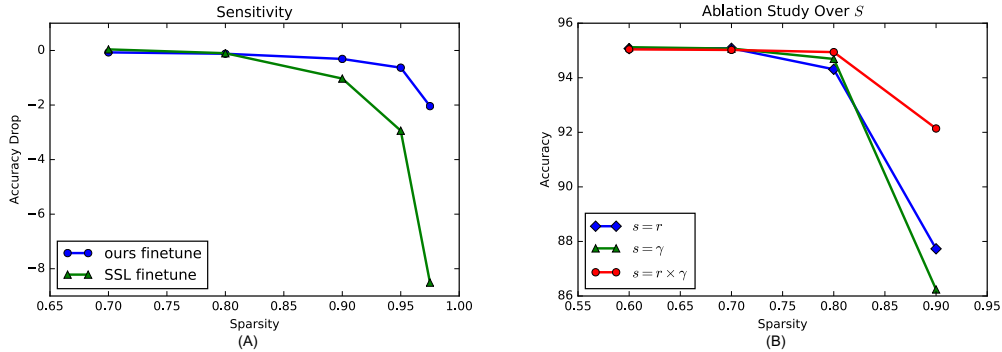

Figure 4: (A) The Accuracy drop-Sparsity curve. Compare to SSL, synaptic pruning is better at preserving accuracy for high pruning rate. Which show that compared with SSL, our Synaptic Strength is more accurate as the indicator to connection importance. (B) Ablation study by modifying the definition of Synaptic Strength. Excluding either component has lead to performance degrade,which shows the optimist of our approach.

Table 4: Compare with winograd-domain sparse models on CIFAR-10

| Model | Error(%) | Density |
|---|---|---|
| VGG-nagadomi base | 6.30 | 100% |
| VGG-nagadomi | 7.40(+1.1) | 25% |
| VGG base | 6.20 | 100% |
| VGG ours | 6.23(+0.03) | 4% |

## 5 Practicability

Convolution operation in CNNs is commonly computed by GEneral Matrix Multiplication(GEMM). The computation routine of GEMM requires lowering weight tensor and input tensor to 2D matrices [3]. Using this method, the weight tensor of kernel sparse layer is transformed to a "strip" sparse metrics, which can be accelerated using block sparse matrix multiplication algorithms on gpu [8].

In another track, Winograd convolution, which is adopted recently in convolutional computation [18], optimizes 2D convolution using Winograd decomposition and take summation over multiple 2D convolutions. Several works have been proposed to prune individual weights directly in Winograd domain since fine-grained level sparsity preserved does not preserve after Winograd transform being applied. However, kernel level sparsity remains unchanged after being transformed into "wino-grad domain". Thus we compared our method with the Winograd direct pruning method proposed by [21] in Table 4. Our method could achieve a significantly higher rate of sparsity (about $8\times$) with almost no drop in accuracy.

## 6 Conclusion

Inspired by the synaptic pruning mechanism inside the human brain, we introduced a new class of parameters called Synaptic Strength. We show that we can achieve high pruning with almost no cost at performance using Synaptic Strength. Further analysis proves that Synaptic Pruning is a better indicator of importance compared with the existing method. We will continue to investigate our method by exploring efficient inference methods for kernel sparse CNNs.

## Footnotes

*This work is done when Zhao Zhong is intern at SenseTime Research

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
