[Reviews · NeurIPS 2018]

Reviewer 1



This paper defines a measure of Synaptic Strength inspired by the human brain, to represent the contributed information from a connection. The authors use this method for pruning CNNs with high compression. Sparsity-induced regularization is performed on the synaptic strength. Experiments on both CIFAR-10 and ImageNet show that performance is not compromised with significant reduction in parameters. The authors further present robustness analysis results. The paper is very well-written and presentation is excellent. Minor comments: - loss it -> lose it - going to pruned -> going to prune - several works has -> have

Reviewer 2



Content: This submission introduces a new framework for compressing neural networks. The main concept is to define the "synaptic strength" of the connection between an input layer and an output feature to the the product of the norms of the kernel of the input layer and the norm of the input layer. A large synaptic strength indicates that a certain input feature plays a substantial role in computing the output feature. The synaptic strength is incorporated into the training procedure by the means of an additional penalty term that encourages a sparse distribution of synaptic strengths. After training all connections with synaptic strength smaller than some threshold are fixed to zero and the network is finetuned. The authors apply their method to different networks on CIFAR (including VGG, ReseNet and DenseNet) and ImageNet (ResNet-50) and show good performance (in terms of accuracy and number of remaining parameters and computations). They also look into how the sparsity after training affects accuracy. Quality: The idea of synaptic strength appears intriguing. I like especially the idea of pruning whole connections between input channels and output layers instead of individual weights and it's very interesing to see in which layers most pruning is happening (Figure 3 left). I have some minor issues with the derivations in section 3.2: * eq (1): shouln't the bias be b^{i+1}_k instead of b_{i+1}_c, i.e. the each output feature (k) has a certain bias, not the input channels (c). The same should be the case for the final summand b_c in eq. (3)-(5). * In eq (3)-(5): the convolution and adding of the bias should happen inside the nonlinearity f, not after applying the nonlinearity. Maybe there are just some parantheses missing? * going from eq (3) to eq (4), the factor gamma is moved from inside the nonlinearity outside of it. This works for nonlinearities like relu and prelu (as long as gamma is positive!), but it won't work for other nonlinearities like sigmoids. This condition should be clarified. The results reported in the paper show that the method manages to decrease the number of parameters while keeping good performance. One other quite recent method that I would like to see compared to in this paper is Louizos et al, "Learning Sparse Neural Networks through L_0 Regularization", ICLR 2018 which reports a performance of less than 4% on CIFAR 10 with WideResNets (compared to 5% in this submission with ResNet). One of the reasons why I would like to see this comparison is that Loizos et al report that besides compressing the network, their method also serves as a regularization method that can improve generalization and I'm wondering whether the same might hold for the proposed method in this submission. Table 1 suggests that might be the case for VGG and ResNet-18 on CIFAR10 but not on ImageNet (Table 2). Also I'm wondering how strong the dependence on the regularization parameter lambda is. How where the used parameters choosen and how much do the results differ if a different parameter is used? The bold numbers in Table 3 indicate that the ResNet-50 version of the submission has with 5.9M parameters the smallest number of remaining parameters, however, 3 out of the 4 comparison models in the table have even less parameters. What exactly is the sparsity in Figure 4 (a) and (b)? The parameter lambda? The threshold tau? Something else? Why did the authors not finetune in the ablation study? If the difference would go away after finetuning, this would change the result of the ablation study, right? Clarity: There are some details missing: What was the threshold for pruning connections? The captions of tables 2 and 4 are not indicating which dataset they are using (ImageNet if I'm not mistaken). Please have your submission proof-read for English style and grammar issues. Originality: The authors make an interesting contribution to the wide field of compressing neural networks. For me the main contribution is to make the connection between a single input feature and a single output feature the target of the compression. I'm not aware that this has been done before -- but my knowledge of this field is like not complete. The authors compare to some other compression techniques but are missing on some recent state-of-the-art methods that show really good performance (see above for one example) Significance: Compressing DNNs is a very important topic for bringing deep learning into everyday applications. The concept of synaptic strength is an very interesting contribution to that field. To help convincing other researchers of this method, I would like to see more comparison with state-of-the art and other potential applications of the method like regularization. Minor comments: In the caption of Figure 1, "left" and "right" are switched. Update after the rebuttal: The authors addressed most of my questions and concerns. With the promised changes incorporated I think the paper will be a better contribution and therefore I am happy to increase my rating from 6 to 7.

Reviewer 3



The authors present a novel implicit pruning methodology that works by calculating importance of filter connections between layers. The paper is written very well and touches an important topic. The methodology seems sound and the experiments make sense given the method. A couple of questions & comments: 1. "Variance which is also a uncertainty measurements could be regarded as a acceptable compromise. " (134) The authors could use the ln(2*pi*e*variance) which would actually be the entropy under a Gaussian approximation. This would result in a higher entropy in the distribution over Synaptic Strenght across the synapses. 2. Experiments: a) Further comparison with CondenseNet and other simple pruning strategies are needed at the level of model inspection. Is there a difference in what connections are classified as important? b) How (or is) the distribution of Synaptic Strength affected by how deep the synapses are in the network? (how uniform/peaked on the lower/higher layers)? c) Is the distribution of Synaptic Strength similar across different task? (CIFAR/Iimagenet)? d) Why only top x% performance is reported and not average performance, or performance after convergence? e) How many repetitions where done on the task?